# Adrenoleukodystrophy Newborn Screening in California Since 2016: Programmatic Outcomes and Follow-Up

**DOI:** 10.3390/ijns7020022

**Published:** 2021-04-17

**Authors:** Jamie Matteson, Stanley Sciortino, Lisa Feuchtbaum, Tracey Bishop, Richard S. Olney, Hao Tang

**Affiliations:** Genetic Disease Screening Program, California Department of Public Health, Richmond, CA 94804, USA; Stanley.Sciortino@cdph.ca.gov (S.S.); Lisa.Feuchtbaum@cdph.ca.gov (L.F.); Tracey.Bishop@cdph.ca.gov (T.B.); Richard.Olney@cdph.ca.gov (R.S.O.); Hao.Tang@cdph.ca.gov (H.T.)

**Keywords:** adrenoleukodystrophy, newborn screening, follow-up, evaluation

## Abstract

X-linked adrenoleukodystrophy (ALD) is a recent addition to the Recommended Uniform Screening Panel, prompting many states to begin screening newborns for the disorder. We provide California’s experience with ALD newborn screening, highlighting the clinical and epidemiological outcomes observed as well as program implementation challenges. In this retrospective cohort study, we examine ALD newborn screening results and clinical outcomes for 1,854,631 newborns whose specimens were received by the California Genetic Disease Screening Program from 16 February 2016 through 15 February 2020. In the first four years of ALD newborn screening in California, 355 newborns screened positive for ALD, including 147 (41%) with an *ABCD1* variant of uncertain significance (VUS) and 95 males diagnosed with ALD. After modifying cutoffs, we observed an ALD birth prevalence of 1 in 14,397 males. Long-term follow-up identified 14 males with signs of adrenal involvement. This study adds to a growing body of literature reporting on outcomes of newborn screening for ALD and offering a glimpse of what other large newborn screening programs can expect when adding ALD to their screening panel.

## 1. Introduction

With an estimated birth prevalence of 1 in 17,000, X-linked adrenoleukodystrophy (ALD, OMIN 300100) is the most common inherited peroxisomal disorder [1,2]. ALD is caused by variants in the ATP binding cassette subfamily D member 1 (*ABCD1*) gene, located on the X chromosome, which can lead to increased concentrations of very long-chain fatty acids (VLCFAs) in plasma, as well as in adrenal and nervous tissues [3,4].

As ALD is X-linked, only males are affected by the most severe form of the disorder. ALD has been characterized in males by three phenotypes which vary in the onset and severity of symptoms: Adrenal insufficiency, cerebral ALD, and adrenomyeloneuropathy (AMN) [5]. These phenotypes cannot be predicted by the *ABCD1* variant, concentrations of VLCFAs, or family history, making it difficult to determine prognosis [6,7]. As the disease progresses, it may evolve from one phenotype to another [8]; some boys, for example, begin life asymptomatic, develop adrenal insufficiency within the first few years of life, then develop signs of cerebral ALD a couple of years later. Alternatively, some boys may never develop the cerebral form and may go on to develop AMN as an adult. While carrier status was once believed to be asymptomatic, a subset of females have been shown to exhibit AMN-like symptoms, which generally manifest after the fourth decade of life [9,10].

In males, the most severe form of the disease is cerebral ALD, where demyelination of the cerebral hemispheres results in progressive declines in behavior, reasoning, cognition, and motor function. Cerebral ALD symptom onset has been observed at any age after 2.5 years old, however, in 31–35% of cases, symptoms are observed before the child reaches age ten [5]. Cerebral ALD progresses rapidly, often leading to death or a vegetative state within two to four years of symptom onset [5]. Hematopoietic stem cell transplantation (HSCT) has been shown to be an effective therapy to halt cerebral demyelination and prevent death if administered early, before neurologic progression [11,12,13]. Regular MRIs and adrenal testing are recommended to monitor for early signs of the disease in order to administer treatment when it is most effective [14,15,16,17,18,19].

Adrenal insufficiency is present in 79–90% [15] of boys with childhood cerebral ALD with a lifetime prevalence for all males with ALD at around 80% [17]. Signs and symptoms of adrenal insufficiency can range from fatigue, weight loss, and hypotonia to life-threatening adrenal failure. Fortunately, adrenal insufficiency can be treated through hormone replacement therapy [20]. The early detection of adrenal insufficiency and cerebral ALD highlight the value that newborn screening brings to treating children with this disorder.

ALD was initially nominated for inclusion on the United States’ Recommended Uniform Screening Panel (RUSP) in 2012 [21]. By 2014, a technique using liquid chromatography-tandem mass spectrometry (LC-MS/MS) analysis of C26:0-lysophosphatidylcholine (C26) was validated as a screening methodology with high sensitivity and throughput, making it ideal for newborn screening [22,23,24,25,26]. Anticipating the addition of ALD to the RUSP, the California legislature amended the Health and Safety Code in September 2014 to require that the Department of Public Health screen newborns for ALD on the day of its adoption to the RUSP [27]. ALD was added to the RUSP on 16 February 2016 [26], and California began ALD newborn screening on 21 September 2016, with retrospective screening of the specimens accessioned on or after 16 February 2016 to comply with the aforementioned Health and Safety Code.

The purpose of this article is to highlight California’s ALD newborn screening experience, including the laboratory methods utilized, clinical and epidemiological outcomes observed, and program implementation challenges. To our knowledge, this report describes the largest (and one of the first) population-based studies on ALD newborn screening. It is also one of the first studies to report long-term follow-up findings up to four years of age.

## 2. Materials and Methods

In California, dried blood spots (DBS) are collected from almost all newborns, except the few whose parents opt out for religious reasons. Otherwise, similar to most other state programs, newborn screening is mandatory and is provided to all newborns between 12 and 48 h after birth.

For ALD newborn screening, specimens are sent to the California Department of Public Health (CDPH) Genetic Disease Laboratory in Richmond, CA for analysis, which utilizes a three-tiered approach. DBS are analyzed in the first tier using flow injection analysis-tandem mass spectrometry (FIA-MS/MS) to measure C26:0-lysophosphatidylcholine (C26) with a cutoff of ≥0.42 µmol/L. The second tier utilizes liquid chromatography-tandem mass spectrometry (LC-MS/MS) to provide a more precise measurement of C26 in the DBS. The second-tier cutoff was initially set at ≥0.15 µmol/L, but was adjusted to ≥0.22 µmol/L in December 2017 to improve screening performance. For the third tier, all second tier positive DBS specimens are shipped to a contracted laboratory for Sanger sequencing of the *ABCD1* gene. Variants are classified in accordance with ACMG guidelines [28]. For the purposes of analysis in this study, variant classifications were collapsed into three categories: No variant (including benign, likely benign and polymorphism), variant of uncertain significance, and pathogenic (including likely pathogenic).

After *ABCD1* sequencing results are available, all screen positive newborns (boys and girls) are referred to one of fourteen metabolic Special Care Centers (SCCs) across California regardless of the sequencing result. The SCCs provide genetic counseling and confirmatory testing as they deem appropriate. Confirmatory testing may include plasma VLCFAs analysis, plasmalogen testing, deletion/duplication analysis of the *ABCD1* gene, or further molecular testing (e.g., *PEX* gene). Parental testing is provided as a complimentary follow-up test by the contract laboratory for newborns with a variant found on *ABCD1* sequencing. Further familial testing is ordered at the discretion of the metabolic specialist, and testing charges are covered by third-party payers, when available.

Every case referred for follow-up is reported to CDPH via an online short-term follow-up data collection tool in our Screening Information System (SIS), which includes information about follow-up services provided to the newborn as part of the initial diagnostic work-up, clinical signs and symptoms, and the initial ALD-related diagnosis (“case resolution”). Additionally, metabolic clinics use a long-term follow-up (LTFU) data collection tool in SIS to provide an annual report containing updated health outcome and clinical data for every male diagnosed with ALD that continues to be followed by each special care clinic.

Newborns with a positive screen and an *ABCD1* pathogenic variant or VUS are confirmed to have ALD if plasma VLCFA interpretation is abnormal, or in some cases, borderline. For all newly confirmed ALD cases, the initial case resolution is reported as *ALD Unspecified*. If ALD or any other known disorder is ruled-out, the case resolution is reported as *No Disorder*. The case resolution will remain as *ALD Unspecified* on the annual LTFU report until the specialists provide more precise diagnoses, at which point the case resolution can change to *Addison’s disease only* (if adrenal insufficiency is reported), or *cerebral ALD*, or *AMN*. As only a few boys to date have exhibited signs meriting a change in diagnosis, we have grouped all ALD diagnoses in this report into one “ALD” category.

For this report, we reviewed California newborn screening and follow-up data retrospectively for specimens accessioned from 16 February 2016 through 15 February 2020. LTFU data was analyzed through 21 May 2020. Descriptive statistics, such as frequencies and percentages, were derived using Microsoft Excel^®^. The distribution analysis of C26 analyte values and the significance tests were performed using SAS software, Version 9.4 of the SAS System for Windows, specifically PROC FREQ, PROC SGPLOT, and PROC NPAR1WAY (copyright © 2021–2012 SAS Institute Inc.; SAS and all other SAS Institute Inc. product or service names are registered trademarks or trademarks of SAS Institute Inc., Cary, NC, USA).

## 3. Results

### 3.1. ALD Newborn Screening Outcomes in California

In the first four years of ALD newborn screening, from 16 February 2016 through 15 February 2020, California screened 1,854,631 newborns for ALD and other peroxisomal disorders (Figure 1), including 945,344 males, 900,917 females, and 8370 infants with no biological sex reported; 77,637, or 4.2% had elevated C26 at the first tier and went onto the second tier of screening; 355 or 0.5% were screen positive on the second tier of screening and had *ABCD1* gene sequencing performed before they were referred to a metabolic clinic for a diagnostic workup. Females made up a higher proportion of screen positives at 57% of cases.

Of the 355 screen positive referrals, 68% (240/355) were resolved with a disorder and 26% (93/355) were considered false positive and classified as “no disorder” (Figure 1). Twenty-two resolutions could not be determined due to inconclusive (*n* = 7) or no confirmatory testing at the time of this study (*n* = 3), parent refusal of confirmatory testing (*n* = 7), or the patient being deceased (*n* = 2) or out of state before tests could be ordered (*n* = 3).

Of the 240 infants resolved with a disorder, 95 (40%) males were diagnosed with ALD, and 110 (46%) females were diagnosed as heterozygous carriers. An additional 23 screen positive cases that had significantly elevated C26, but were absent an *ABCD1* variant were identified with Zellweger Spectrum Disorder (ZSD). A small proportion of cases were classified as having another unspecified condition.

In our screening population, the mean C26 was 0.27 µmol/L in the first tier of screening and 0.075 µmol/L in the second tier (Figure 2). In both tiers, mean C26 was similar across groups when analyzed by sex, race/ethnicity, birthweight, gestational age, age at collection, and nursery type. Males diagnosed with ALD had significantly higher C26 means than screen positive newborns resolved with no disorder at both the first (0.79 µmol/L vs. 0.51 µmol/L) and second tiers (0.42 µmol/L vs. 0.21 µmol/L), although there was substantial overlap in the range of values between the groups. Newborns identified with ZSD had C26 means that were considerably greater than the other resolution groups, with a mean C26 of 2.26 µmol/L at the first tier of screening and 1.43 µmol/L at the second tier.

### 3.2. ABCD1 Sequencing and Short-Term Follow-Up

Among the 355 cases referred, SCCs reported that plasma VLCFA confirmatory testing was performed for 264 (74%), with males being more likely to have the test performed (84% or 130/154), compared to females (67% or 134/201) (*p* < 0.0001). Follow-up centers reported “abnormal” plasma VLCFA results in 189 of the 260 cases (73%) where results were provided. In a previous study, we found that an analytical tool (CLIR) based on confirmatory testing markers produced results correlating very well with genotypes [29,30].

In four years of ALD screening, we identified 149 different *ABCD1* variants (Appendix A). The most common pathogenic variants were NM_000033.3:c.565C>T (10 cases) and NM_000033.3:c.1166G>A (7 cases). The most common VUS were NM_000033.3:c.839G>A (8 cases) and NM_000033.3:c.1448C>T (6 cases). All other variants were observed in five or fewer cases, including 110 variants that were only observed in one case.

Of the 355 cases referred for *ABCD1* sequencing and diagnostic follow-up, we found that 41% (147/355) were determined to have a variant of uncertain significance (VUS), followed by 35% (124/355) with no identified variant, and 24% (84/355) with a pathogenic variant (Table 1). Cases with a pathogenic variant had a diagnostic resolution reported by the SCC specialist in fewer days (*Mean* = 55, *Standard Deviation* = 72, *Median* = 29.5, *Interquartile Range* = 7–63) than cases with no variant (*Mean* = 90, *Standard Deviation* = 97, *Median* = 53, *Interquartile Range* = 22–114) or a VUS (*Mean* = 96, *SD* = 98, *Median* = 54, *Interquartile Range* = 19–160) (*p* = 0.014). Male cases were resolved in fewer days (*Mean* = 73, *SD* = 83, *Median* = 36, *Interquartile Range* = 15–110) than females (*Mean* = 92, *Standard Deviation* = 99, *Median* = 50.5, *Interquartile Range* = 20–134.5) (*p* = 0.043).

All of the males with pathogenic variants (33/33) and 86% (60/70) of males with a VUS were diagnosed with ALD. 10% (7/70) of males with a VUS were resolved as no disorder, and 4% (3/70) could not be determined. A majority (63% or 33/52) of males with no variants were resolved as no disorder, while a substantial proportion (13% or 7/52) were diagnosed with ZSD, and 3 cases (6%) were diagnosed with ALD. Of the males that were diagnosed with ALD where no *ABCD1* variant was found on Sanger sequencing, one had an *ABCD1* gene deletion discovered through confirmatory testing, one had a mosaic variant discovered during confirmatory testing, and 1 was suspected to have a variant not detectable by testing.

Among the 77 females with a VUS, 70% (54/77) were diagnosed as heterozygous ALD carriers, while 16% (12/77) were resolved with no disorder, and 10% (8/77) could not be determined. A majority (57%, or 41/72) of females with no reported variants were resolved with no disorder, while 21% (15/72) were diagnosed with ZSD. Of the four female cases (6%, or 4/72) with no *ABCD1* variants, three were found to have an *ABCD1* gene deletion discovered through confirmatory testing; only one did not have further *ABCD1* molecular testing performed.

### 3.3. ALD Prevalence and Screening Performance in California

876,131 specimens were screened for ALD between 16 February 2016, and 12 December 2017, before CDPH adjusted the second-tier cutoff. During that time, we observed a birth prevalence of 1 in 7181 for males and 1 in 7424 for heterozygous females (Table 2). After changing the cutoff, we screened 978,500 specimens for ALD and observed a birth prevalence of 1 in 14,390 for males and 1 in 9593 for heterozygous females. The screen positive referral rate declined from 1 in 4056 to 1 in 7040 after changing the second-tier cutoff. After the cutoff change, we also observed a reduction in the proportion of patients with a VUS, from 47% of all positive cases to 32%

### 3.4. Long-Term Follow-Up (LTFU) Outcomes

Of the 95 males with ALD that will be followed by CDPH for LTFU data, 82 were at least one year of age, thus prompting metabolic SCCs to complete an annual survey on the status of the child in the previous year of care. At the time of data analysis, LTFU data was submitted for 71 of the 82 cases (87%), with 63 reports for the first year of follow-up, 52 reports for the second year of follow-up, 15 for the third year of follow-up, and 3 for the fourth year of follow-up (Table 3).

None of the 95 boys had signs or symptoms of ALD reported at the time of diagnosis. In just over three years of collecting LTFU data, 14 patients were reported to have adrenal involvement. By age one, 8 patients had abnormal adrenocorticotropic hormone (ACTH) tests, two of which had adrenal insufficiency and treatment with hydrocortisone (or other glucocorticoid). By age two, 7 patients had abnormal ACTH tests, including one with poor weight gain that was treated with hydrocortisone (or other glucocorticoid) and two more otherwise asymptomatic boys that were treated with hydrocortisone (or other glucocorticoid). By age 3, one patient had an abnormal ACTH test, poor weight gain, and was treated with hydrocortisone (or other glucocorticoid). No adrenal involvement was reported for the 3 patients that reached four years of age. The youngest boy with adrenal involvement was 5 months old, with treatment beginning at almost 6 months.

Of the 71 cases for which LTFU data was submitted, 23 patients had at least one MRI reported. The median age that the first MRI was performed was 21 months, with first MRIs performed as early as 3 months and as late as 35 months. Abnormal MRI findings were reported for one patient at 18 months of age who was also reported to have behavioral changes, seizures, and other neurological findings; one patient with adrenal insufficiency had three serial MRIs at 25, 29, and 35 months of age with abnormal MRI findings; another patient with abnormal MRI findings at 21 months was reported to have experienced seizures; and finally, two asymptomatic patients had abnormal MRI findings, which were reported at 22 months for one patient and 47 months for the other. No patients have been reported to have undergone a hematopoietic stem cell transplant.

## 4. Discussion

After our adjusted second tier cutoff, California observed an ALD birth prevalence of 1 in 14,397 males. This compares to a birth prevalence of 1 in 18,783 found in New York state [16], 1 in 3878 in Minnesota [31], 1 in 8717 in South Carolina [32], and 1 in 51,081 in Georgia [33]. Differences between screening cutoffs, case definitions, and *ABCD1* carrier frequencies in our population samples are likely leading to the reported variations in birth prevalence between states.

In California, the observed birth prevalence of ALD decreased markedly after we adjusted the second-tier cutoff. This difference is likely due to an excess in *ABCD1* VUS identified before the cutoff changed. For those unclear cases, clinicians might have diagnosed cases conservatively with ALD in order to monitor the children for symptoms. As a result, we might find that some of the boys who are currently diagnosed with ALD are actually false-positive resolutions, which would decrease our birth prevalence rates further. In fact, an analysis of plasma VLCFA using the CLIR post-analytical tool reduced the number of diagnosed cases among the VUS by more than 60% [29]. In our data, birth prevalence rates for heterozygous females decreased much less dramatically than males after the second-tier cutoff change. Further investigation is warranted, as an explanation for this phenomenon is unclear.

We believe that having false negatives for males in our population is less likely, as the methodology has been proven to be quite sensitive. In the first four years of newborn screening, only one possible false negative was reported. The child was found to have the same VUS as his brother who was detected through newborn screening; however, he did not have elevated C26 when tested on plasma. As the child is too young to have symptoms of cerebral ALD, it is too early to know if it is truly a missed case. Additional missed cases may arise as our cohort becomes old enough to exhibit symptoms and present clinically.

Alternatively, it is possible that some cases which were screen positive may have been misdiagnosed due to a lack of genetic information. The Sanger sequencing provided by our program does not identify small deletions in the *ABCD1* gene which make up approximately 3–6% of ALD cases [7,34,35]. In our population, we found that 2% (4/205) of infants diagnosed with ALD or as heterozygous females had an *ABCD1* deletion which was not detected on Sanger sequencing. We might, therefore, have an underestimation of cases with deletions, especially in females where VLCFA testing was less likely to be performed.

As a secondary finding of ALD newborn screening, we identified 1 in 80,657 California newborns with ZSD, slightly lower than the expected 1 in 50,000 [36]. Though the rate increased considerably after we changed the second tier cutoff, this is most likely due to the rarity of the disorder in a small sample size, as our cutoffs were well below the range of C26 values in newborns who were found to have ZSD. Since ZSD is not the primary target of ALD newborn screening, follow-up centers are not required to report clinical diagnoses, and we cannot evaluate the sensitivity of our screening methodology for ZSD.

We observed several early indicators of symptomatic ALD in the group of 95 boys that we collected LTFU data on. Adrenal involvement was observed in a boy as young as 5 months old. Five boys were reported to have undergone treatment for adrenal symptoms, an important step towards preventing a potentially life-threatening condition. Abnormal MRIs were noted in several boys, however HSCT had not been reported, leading us to question whether the abnormal MRI findings are related to a cerebral ALD diagnosis.

In our dataset, only 23 out of 71 patients had an MRI reported. This was lower than expected, given the MRI monitoring recommendations. Upon further investigation, we found a couple of factors which may have contributed to the low rate of MRIs. First, 23 of the patients with LTFU data reported only had data submitted for the first year of life, which is younger than the age when an MRI is recommended. We also believe that some of our earlier data on MRIs may be incomplete as the question on the form was optional. Despite this limitation, we believe the question at minimum elicited data on abnormal MRIs, which is most important in determining whether we detected any cases of cerebral ALD. We have since modified our data collection form to explicitly ask this question, as this is an essential measure of success for newborn screening.

Our analysis of LTFU data is preliminary, as data collection is only in the very early stages for ALD. For most disorders, we collect LTFU data annually for five years, but for ALD, we will extend the years of LTFU to 21 years of age. This extended surveillance will be important in some cases where signs of cerebral ALD do not develop until age ten or older. Our LTFU data collection process gives us the unique opportunity to refine our birth prevalence estimates, adjust our cutoffs if necessary, learn more about disease-causing variants, and the overall post-screening natural history of ALD.

In the first four years of screening, we identified as many as 95 California boys affected with ALD, thus providing the potential for lifesaving treatment with ongoing monitoring by MRIs and adrenal testing. Despite these achievements, implementing ALD newborn screening did not come without its challenges.

ALD was the first newborn screening disorder that was required under a timeline mandated by the California Health and Safety Code to commence screening the day ALD was added to the RUSP. Given the complexity of screening approximately 1800 newborns a day, we consider it an achievement to bring up newborn screening only seven months after ALD was signed on to the RUSP. Unfortunately, this still did not meet the mandated timeline, so CDPH was required to retroactively screen nearly 300,000 specimens. A revised legislative mandate now allows CDPH to go-live with new disorders within 2 years after a new disorder is added to the RUSP.

ALD screening also posed a unique set of challenges due to the X-linked nature of the disorder. For females identified with an *ABCD1* variant, ALD may manifest as an adult-onset disorder when symptoms arise, but a large proportion of females are expected to be asymptomatic carriers. Providing molecular results for female newborns raises ethical questions, since we give them no choice to test for a potential adult-onset disorder. After consultation with specialty care centers, we decided that we would provide molecular results at the time of referral for all newborns, regardless of sex to give clinicians the opportunity to counsel patients and to follow-up on screen positive results. We found that females made up a larger proportion of the referrals and took longer to resolve than males. Protracted uncertainty has the potential to heighten anxiety for parents of females who receive screen positive results for a disease that may never manifest [37]. A recent study of family perspectives on California ALD newborn screening found that parents were in favor of including ALD on the newborn screening panel for both males and females despite the anxiety associated with receiving a screen positive result [38].

An ALD diagnosis in a female newborn often leads to a cascade of testing multiple family members to find other affected males and could require counseling all females of reproductive age. On the other hand, identification of female carriers can be especially beneficial to young male siblings, as well as mothers who may be considering future children. This benefit, however, comes with the additional cost of familial testing, which is typically not covered by newborn screening programs, and may not be covered by third-party payers. Fortunately, our sequencing laboratory offered to provide parental testing as a complimentary service.

Another significant challenge that we faced was the high proportion of cases that were found to have a VUS. This fact, compounded by poor genotype–phenotype correlation, makes it difficult for clinicians to arrive at a resolution on how to diagnose these cases, often requiring more tests to be ordered and more time to diagnose. While we expected a large proportion of VUS due to the number of novel *ABCD1* variants documented in the literature [31,34,39], we identified even more than was expected. To classify variants more precisely, we worked with our contract laboratory to reference a well-known ALD variant database [40].

Third-party payers were initially reluctant to cover serial, frequent MRIs and adrenal testing recommended among asymptomatic males, despite the fact that these monitoring services are so important to diagnose the advancement of disease; this was especially difficult with the cases that presented with a VUS. CDPH connected the third-party payers with California specialists who advocated for the frequent monitoring services, and in the end, the coverage was expanded more widely to all males identified with ALD, regardless of the variant type. CDPH also addressed this issue by modifying the tier 2 C26 cutoff, which reduced the proportion of screen positive patients with a VUS.

As we began to collect LTFU data, we were met with another challenge: Determining where the specialty “medical home” would be for these patients. While the initial newborn screening referral goes to geneticists at the metabolic SCCs, these patients are also seen by endocrinologists and neurologists once the ALD diagnosis is made. It is important for us to gather information about visits from all specialists in order to properly evaluate ALD newborn screening. Our current system allows for a transfer to be made from the geneticist to the endocrinologist, but we might need to develop more flexibility in our data system to allow multiple specialists to furnish information about the patient when they provide ongoing care for these children across disciplines and over time.

Our analysis showed that, on average, the first MRIs were performed at 21 months of age, later than the current recommendation to have the first MRI performed at 12–18 months of age. This discrepancy exemplifies the challenge that our program and our partnering SCCs faced when trying to enforce a uniform approach to monitor these cases when more formal consensus statements were yet to be published. Recommendations were eventually published, but the publication dates were well after screening had begun [14,15,19]. In the interim, our program worked collaboratively with specialists across California and consulted with other state screening programs [25] to provide draft guidelines until a national consensus could be reached.

## 5. Conclusions

We provided outcomes on the largest cohort of newborns screened for ALD to date, screening more than 1.8 million newborns in California and identifying 95 males with ALD in the first four years of screening. This study adds to a growing body of literature which shows that ALD prevalence rates vary between screening programs, likely due to differences in screening methodology, case definitions, and *ABCD1* carrier frequencies in our population samples. We have also seen how cutoff adjustments can potentially improve screening test performance, thus reducing unnecessary referrals and parental anxiety. However, without a robust LTFU system in place to monitor the health status of identified cases over time, it will be difficult to know if newborn screening has been successful in helping to prevent the otherwise high morbidity and mortality rates associated with ALD.

For now, the California Newborn Screening Program will continue monitoring cases through our annual patient assessments with the hope that the information garnered over time will expand our knowledge about ALD genotype/phenotype associations and will lead to improved disease management, health outcomes, and ultimately, to be able to adequately assess the effectiveness of newborn screening for this neurodegenerative disease.

## Figures and Tables

**Figure 1 IJNS-07-00022-f001:**
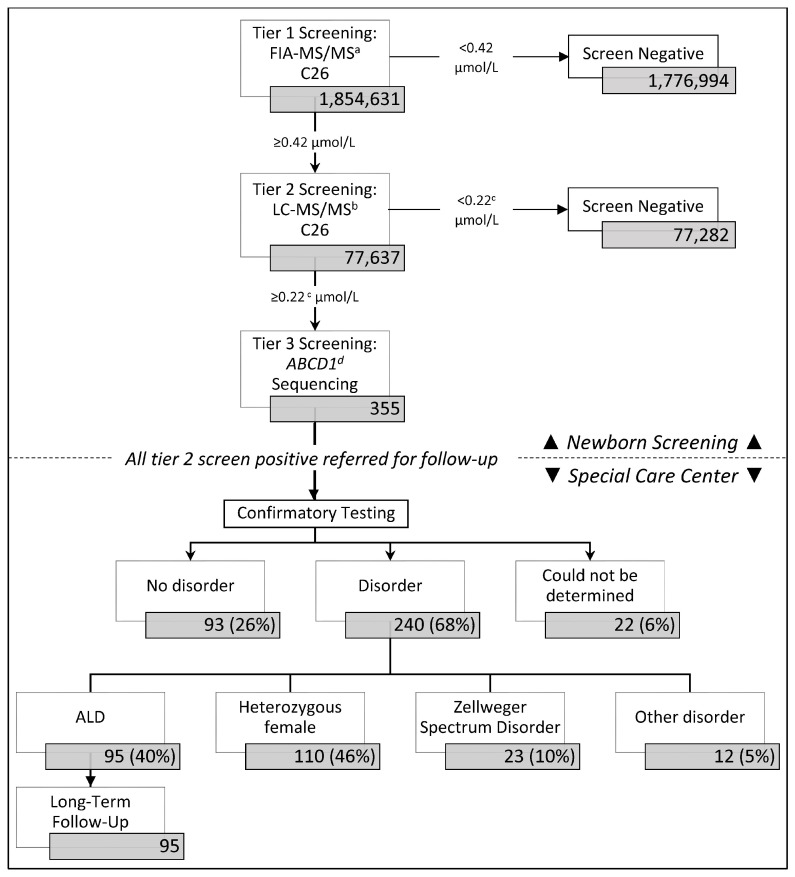
California adrenoleukodystrophy (ALD) newborn screening and follow-up algorithm. ^a^ FIA-MS/MS = flow injection analysis-tandem mass spectrometry. ^b^ LC-MS/M = liquid chromatography-tandem mass spectrometry. ^c^ The 2nd tier cutoff was originally set at ≥0.15 µmol/L, but was adjusted to ≥0.22 µmol/L in December 2017 to improve screening performance. ^d^
*ABCD1* = ATP binding cassette subfamily D member 1 gene.

**Figure 2 IJNS-07-00022-f002:**
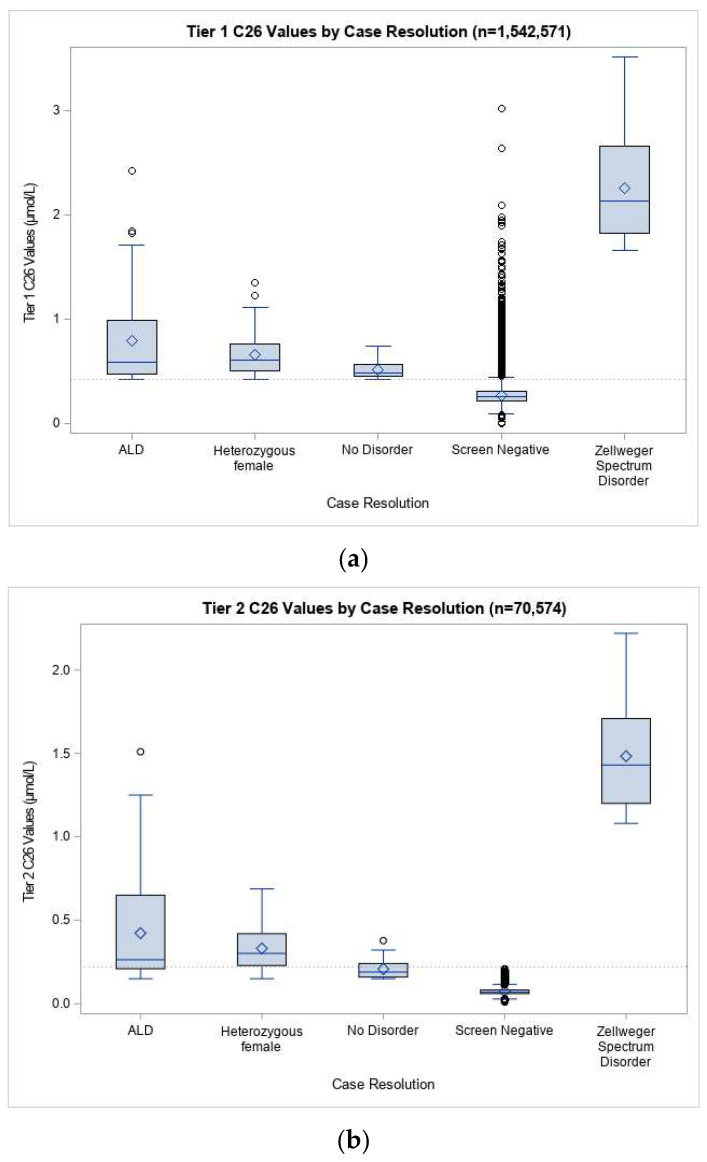
Distribution of C26 values at (**a**) first and (**b**) second tiers of ALD screening for specimens received 21 September 2016 through 15 February 2020. Screen positive data was included in this analysis for specimens received from 16 February 2016 through 15 February 2020. Screen negative data were not available in our Screening Information System before the system went live on 21 September 2016, and were therefore not included in this analysis. Diamond symbols represent the mean. Open circles are outliers beyond 1.5 times the interquartile range.

**Table 1 IJNS-07-00022-t001:** Short term follow-up outcomes for screen positives by sex and *ABCD1* variant classification.

	Males				Females				
	Pathogenic	VUS	No Variant	All Males	Pathogenic	VUS	No Variant	All Females	All Cases
Screen Positive	32	70	52	154	52	77	72	201	355
Cases with Plasma VLCFA ^a^ Performed	27	59	44	130	27	53	54	134	264
Resolved	32	70	52	154	52	77	72	201	355
ALD (Male)	32	60	3 ^b^	95	-	-	-	-	95
ALD heterozygous female	-	-	-	-	52	54	4 ^c^	110	110
No disorder	0	7	33	40	0	12	41	53	92
Zellweger Spectrum Disorder	0	0	7	7	0	1	15	16	23
Other disorder	0	0	6	6	0	2	4	6	12
Could not be determined	0	3	3	6	0	8	8	16	22

^a^ VLCFA = Very Long Chain Fatty Acid confirmatory testing. ^b^ Through confirmatory testing, 1 case was found to have an *ABCD1* deletion, 1 had a mosaic variant, and 1 was suspected to have a variant undetectable by exon sequencing. ^c^ Through confirmatory testing, 3 cases were found to have an *ABCD1* deletion, and one did not have further molecular testing.

**Table 2 IJNS-07-00022-t002:** Observed birth prevalence before and after the second-tier cutoff change on 12 December 2017.

	Before Cutoff Change			After Cutoff Change		
	(*n* = 876,131)			(*n* = 978,500)		
Resolution	Count	Birth Prevalence		Count	Birth Prevalence	
		(1 in)	(per 1,000,000)		(1 in)	(per 1,000,000)
ALD (male)	61	7181	139	34	14,390	69
ALD heterozygous female	59	7424	135	51	9593	104
Zellweger Spectrum Disorder	6	146,022	7	17	57,559	17
Other disorder	6	146,022	7	6	163,083	6
All disorders	132	6637	151	108	9060	110

**Table 3 IJNS-07-00022-t003:** Long term follow-up ALD clinical findings, 2017–2020.

	Year 1	Year 2	Year 3+	Total Cases ^a^
Reports with Signs of Adrenal Involvement (% of total reports)				
Abnormal ACTH test	8 (13%)	7 (13%)	1 (6%)	14 (20%)
Poor weight gain, hyperpigmentation, or adrenal insufficiency	2 (3%)	1 (2%)	1 (6%)	4 (6%)
Treated with hydrocortisone (or other glucocorticoid)	2 (3%)	3 (6%)	1 (6%)	5 (7%)
Reports with Signs of Cerebral Involvement (% of total reports)				
Abnormal MRI	0 (0%)	3 (6%)	2 (11%)	5 (7%)
Behavior changes, seizures, or other neurologic problems	0 (0%)	1 (2%)	1 (6%)	1 (1%)
Treated with HSCT	0 (0%)	0 (0%)	0 (0%)	0 (0%)
Total Reports Submitted	63	52	18	71

^a^ Total cases may be less than the sum of the three columns if signs of adrenal or cerebral involvement were reported in more than one year of follow-up for the same child.

## Data Availability

Deidentified datasets are available through a data request review and approval process from the California Biobank Program in accordance with California Health and Safety Code, Sections 124980(j), 124991(b), (g), (h), and 103850(a) and (d).

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
