# Peer review of "Adrenoleukodystrophy Newborn Screening in California Since 2016: Programmatic Outcomes and Follow-Up"

_2409-515X, 2021, doi:10.3390/ijns7020022_

Round 1

Reviewer 1 Report

The authors describe the newborn screening results and clinical outcomes for 1,854,631 newborns screened for ALD in California between February 2016 to February 2020.

They report that 355 screened positive for ALD giving a birth prevalence in males of 1:14,397 from these while 14 showed evidence of adrenal involvement, none were diagnosed with cerebral ALD.

The paper is timely and as the authors suggest represents one of the first and largest population-based study of newborn screening for ALD.

The paper is well written and presented with good use of English – it is topical and would be valuable to an audience considering the inclusion of ALD screening in their panel.

There are some clarifications needed to help the reader understand the data and the conclusions.  

It seems clear from p2 line 92/93 that an elevated C26 on second tier testing by LC-MS/MS is used to trigger clinical referral.   This is not clear in Figure 1 on p4.   It would appear from the flow chart, that Tier 3 , genomic screening, plays a role in the determination to refer a patient for clinical follow-up and I do not believe, from the written description, that this is the case.   Figure 1 needs to be re-drawn, perhaps showing that, starting with 1,854,631 babies, 1,776,994 are screen negative; 77637 go on to Tier 2 testing, of which 355 are above the cut-off and are referred, and that 77,282 are below the cut-off and are reported as screen negative.   The Tier 3 genomic testing is then undertaken in all 355 babies but is not used to determine clinical referral.

Of the 355 babies who were referred for follow-up, 95 boys were referred to as having ‘ALD’ and required long term follow-up, unfortunately, the case definition to confer the diagnosis as ‘ALD’ is not explained and this a serious omission.    As the authors point out on p8, line 287, differences in case definition is a contributory factor which makes it difficult to compare and understand the observed variation in birth prevalence of ALD in the US.    It is therefore particularly important in this large series that the authors describe the case definition which underlies the label of ‘ALD’ in the boys described.

With these two changes – the redesign of Figure 1 and the inclusion of a case definition for the label of ‘ALD’ then I would recommend that the paper is published.

Reviewer 2 Report

This is an important paper on first ALD newborn screening in California, with a high number of newborns screened and important implications. While the paper is intelligently written and shows no flaws in the English language, the disease background of adrenoleukodystrophy is not well understood and hence several misleading statements are being made.

  1. it is difficult to state that birth prevalence rates vary widely across NBS programs when the CA prevalence lies in between that of Minnesota and NY. Additional discrepancies may be more due to differences in cutoffs chosen rather than true birth prevalence rates. This should therefore be de-emphasized.
  2. not surprising that no CA boys were diagnosed with CALD as the high risk period is from 4-10 years of age and only historically affects 30%. Hence you would not expect to have diagnosed any boys after 4 years of screening. This statement should be removed from abstract as entirely consistent with prior historical knowledge. 
  3. the value of adrenal insufficiency detection is not appropriately acknowledged. This is a life threatening condition and detecting this with NBS is likely the greatest value of early detection as also treatable.
  4. authors state that 355 newborns screened positive for ALD. Actually since this only included VLCFA screening, it is not screening for ALD alone but for peroxisomal disorders overall. 
  5. further, the followup testing by SCC revealed that only 73% had truly elevated VLCFA. Hence the denominator of 355 is misleading. ABCD1 VUS among patients with normal VLCFA is meaningless. 
